# Peptidases Are Potential Targets of Copper(II)-1,10-Phenanthroline-5,6-dione Complex, a Promising and Potent New Drug against *Trichomonas vaginalis*

**DOI:** 10.3390/pathogens12050745

**Published:** 2023-05-22

**Authors:** Graziela Vargas Rigo, Fernanda Gomes Cardoso, Matheus Mendonça Pereira, Michael Devereux, Malachy McCann, André L. S. Santos, Tiana Tasca

**Affiliations:** 1Faculdade de Farmácia and Centro de Biotecnologia, Universidade Federal do Rio Grande do Sul, Porto Alegre 90610-000, RS, Brazil; grazivrigo@gmail.com (G.V.R.); fe12gomes@outlook.com (F.G.C.); 2CIEPQPF, Department of Chemical Engineering, University of Coimbra, Rua Sílvio Lima, Pólo II—Pinhal de Marrocos, 3030-790 Coimbra, Portugal; matheus@eq.uc.pt; 3The Inorganic Pharmaceutical and Biomimetic Research Centre, Focas Research Institute, Dublin Institute of Technology, D08 CKP1 Dublin, Ireland; michael.devereux@dit.ie; 4Chemistry Department, Maynooth University, National University of Ireland, W23 F2H6 Maynooth, Ireland; malachy.mccann@mu.ie; 5Laboratório de Estudos Avançados de Microrganismos Emergentes e Resistentes (LEAMER), Departamento de Microbiologia Geral, Instituto de Microbiologia Paulo de Góes, Universidade Federal do Rio de Janeiro, Rio de Janeiro 21941-902, RJ, Brazil; andre@micro.ufrj.br

**Keywords:** *Trichomonas vaginalis*, metallopeptidases, cytolysis, proteolytic activity, copper-phendione

## Abstract

*Trichomonas vaginalis* is responsible for 156 million new cases per year worldwide. When present asymptomatically, the parasite can lead to serious complications, such as development of cervical and prostate cancer. As infection increases the acquisition and transmission of HIV, the control of trichomoniasis represents an important niche for the discovery and development of new antiparasitic molecules. This urogenital parasite synthesizes several molecules that allow the establishment and pathogenesis of infection. Among them, peptidases occupy key roles as virulence factors, and the inhibition of these enzymes has become an important mechanism for modulating pathogenesis. Based on these premises, our group recently reported the potent anti-*T. vaginalis* action of the metal-based complex [Cu(phendione)_3_](ClO_4_)_2_.4H_2_O (Cu-phendione). In the present study, we evaluated the influence of Cu-phendione on the modulation of proteolytic activities produced by *T. vaginalis* by biochemical and molecular approaches. Cu-phendione showed strong inhibitory potential against *T. vaginalis* peptidases, especially cysteine- and metallo-type peptidases. The latter revealed a more prominent effect at both the post-transcriptional and post-translational levels. Molecular Docking analysis confirmed the interaction of Cu-phendione, with high binding energy (−9.7 and −10.7 kcal·mol^−1^, respectively) at the active site of both TvMP50 and TvGP63 metallopeptidases. In addition, Cu-phendione significantly reduced trophozoite-mediated cytolysis in human vaginal (HMVII) and monkey kidney (VERO) epithelial cell lineages. These results highlight the antiparasitic potential of Cu-phendione by interaction with important *T. vaginalis* virulence factors.

## 1. Introduction

Trichomoniasis is among the most common non-viral sexually transmitted infectious diseases. It is caused by the widespread obligate extracellular parasitic protozoan *Trichomonas vaginalis*. According to the World Health Organization (WHO), approximately 156 million new cases of trichomoniasis are reported worldwide per year [1]. Although many cases remain asymptomatic, patient complaints include the presence of itching, discharge, and local inflammation [2]. In addition, the presence of the parasite can lead to serious complications generated by the production of key molecules (e.g., different classes of extracellular hydrolytic enzymes) that allow the establishment and pathogenesis of infection. For example, *T. vaginalis* genome sequencing has demonstrated the presence of 440 peptidases’ coding genes responsible for composing the degradome of *T. vaginalis*, acting in both physiological and pathological events that culminate in a successful infectious process. In general, peptidases are hydrolytic enzymes able to cleave peptide bonds in peptides/proteins, and are classified depending on the amino acid/metal present in their catalytic site that is responsible for binding and cleaving proteinaceous substrates [3].

*Trichomonas vaginalis* peptidases are classified as cysteine (total of 220 genes), metallo (123 genes), serine (80 genes), threonine (17 genes), and aspartic (6 genes) peptidases. Metallo (MP) and cysteine (CP) peptidases are responsible for more than half of proteolytic diversity in *T. vaginalis* [4]. CPs have about seven clans composed of fourteen families, including papain (C1), calpain (C2), ubiquitin hydrolases (C12 and C19), and metacaspases (C14), among others [4]. Importantly, representatives of CPs are responsible for the degradation of secretory leukocyte protease inhibitor, a substance involved in human immunodeficiency virus (HIV) protection by inhibiting the entry of the virus into monocytic cells, making trichomoniasis a facilitating agent for HIV acquisition [5]. MPs are hydrolases that require the presence of a metal to perform the cleavage of peptide bonds in the target substrates. In *T. vaginalis*, eight clans containing fourteen families have been described, including leishmanolisin (or gp63) peptidases (M8), FtsH endopeptidases (M41), and several families of aminopeptidases, such as P aminopeptidases (M24) [4]. In general, peptidases are ubiquitously found in living organisms and are responsible for several vital functions in microbial cells, including nutrition, proliferation, growth, differentiation, and signaling, as well as pathological functions such as adhesion events, escape, dissemination, and immunomodulation [6,7]. Moreover, the microhabitat has been shown to play a potential role in the selection of certain functions related to peptidases. Different peptidases in *T. vaginalis* have been identified in the parasite secretome, such as subtilisin serine peptidases (S8), papain-type CPs, and MPs of the families M8 and M24 [8].

According to Arroyo [9], 123 genes encoding MPs have been described in *T. vaginalis*. TvMP50 and TvGP63 are most known MPs in trichomonads, both contributing to host cell damage. TvMP50 is an immunogenic MP detected during trichomoniasis in serum samples from male patients. The activity of this enzyme is mediated by Zn^2+^, which is present in high levels in the male urogenital microenvironment, being considered part of the group of cytolytic effectors involved in the pathogenicity of *T. vaginalis* mediated by environmental conditions [10,11]. TvGP63 is a family of enzymes that has 48 members, of which 37 members are transmembrane proteins; its activity has been related to cytotoxicity to HeLa cell monolayers [12,13]. Considering the participation of peptidases in the pathogenesis and virulence of trichomoniasis, these enzymes stand out as promising targets for the development of new anti-*T. vaginalis* therapies.

The compound 1,10-phenanthroline-5,6-dione (phendione) has been the focus of several biological studies due to its favorable chemical and pharmacological versatility [14,15,16]. The antibacterial effects of phendione and its silver and copper derivatives have been reported against the widespread multidrug-resistant gram-negative bacterium *Pseudomonas aeruginosa*, revealing that elastase B (a metallo-type peptidase that is a well-known virulence factor of this pathogen) is a target of these metal-based complexes [14,17]. In addition, these complexes have displayed antiparasitic activity against protozoa such as *Leishmania amazonensis*, *Leishmania braziliensis*, *Leishmania chagasi*, and *T. vaginalis* [16,18,19]. Interestingly, the interaction of copper(II) and silver(I)-phendione compounds with MPs have been described as key target in its antiparasitic activity [16]. In this context, phendione-based compounds were able to inhibit the main metallopeptidase activity (gp63) produced by *L. brasiliensis* promastigotes, affecting either the infection establishment or infection maintenance processes between parasites and macrophages [16]. Moreover, Cu-phendione induced several impacts on metabolic activity and membrane potential parameters of these *Leishmania* species [19]. Similarly, our group previously demonstrated the potent action of phendione and its silver and copper complexes against *T. vaginalis,* especially the copper derivative (Cu-phendione), which presented values of minimum inhibitory concentration comparable to metronidazole, a common drug used in clinical therapy [18]. Furthermore, Cu-phendione and metronidazole acted synergistically in order to more effectively kill *T. vaginalis* trophozoites [18]. In addition, we showed that Cu-phendione killed *T. vaginalis* through redox homeostasis imbalance and apoptosis, a mode of action that is quite distinct from that caused by metronidazole [20].

Considering the promising contribution of phendione and its silver and copper derivatives in the prospection of therapeutic alternatives against trichomoniasis, as well as the interaction of these compounds with MPs produced by clinically relevant parasites, in the present study we investigated the effect of phendione and its metal-based complexes against *T. vaginalis* peptidases. Through evaluation of enzyme function and gene expression, significant disturbance in the proteolytic pathway was observed after parasite treatment with Cu-phendione. These data were corroborated by in silico approaches using TvMP50 and TvGP63 as *T. vaginalis* MP models. Finally, a cytoprotective effect was observed during the interaction process between *T. vaginalis* and both vaginal and renal epithelial cells induced by Cu-phendione treatment.

## 2. Materials and Methods

### 2.1. Parasites

The *T. vaginalis* isolate ATCC 30236 was used in this study, and was cultivated in trypticase-yeast extract-maltose (TYM) medium, pH 6.0, supplemented with 10% adult bovine serum and maintained at 37 °C [21]. Organisms in the logarithmic phase of growth exhibiting normal morphology and motility were counted in a hemocytometer using trypan blue exclusion dye (0.2%), then the parasitic density was adjusted to 2 × 10^5^ trophozoites/mL. All experiments were conducted in triplicate with at least three independent cultures (*n* = 3). To evaluate the cytotoxicity of parasites against mammalian cells, the fresh clinical isolates TV-LACM15, TV-LACM22, and TV-LACH4 were cultivated as described above. The fresh clinical isolates were obtained from urine and vaginal discharge and maintained under cryopreservation (UFRGS Research Ethical Committee approved the assays under authorization number 18923).

### 2.2. Treatment

Cu-phendione was prepared in accordance with the methods previously described by McCann et al. [22]. Treatment occurred by exposing trophozoites (2 × 10^5^ trophozoites/mL) to Cu-phendione at MIC value for 2 h. The minimal inhibitory concentration (MIC) value for TV-ATCC 30236 and TV-LACH4 was 12.5 µM, while for TV-LACM15 and TV-LACM22 it was 6.25 µM [18]. For all analyses, control condition refers to trophozoites grown in culture medium without Cu-phendione.

### 2.3. Computational Analysis of TvMP50, TvGP63, and TvCP2

The amino acid sequences of TvMP50 (TVAG_403460) (MEROPS: MER0082185; uniprot: A2F8Y2; genbank: EAX98664) and TvGP63 (TVAG_367130) (MEROPS: MER0078144, uniprot: A2G9D5, genbank: EAX86230.1) were submitted to Alphafold2 software for prediction of three-dimensional structure [23]. The TvCP2 predicted structure was obtained from the AlphaFold Protein Structure Database (uniprot: Q27107). Subsequently, the protonation states of the titrable residues of TvMP50, TvGP63, and TvCP2 were calculated using ProteinPrepare (PlayMolecule web server—playmolecule.org, accessed on 11 November 2022) [24]. The MP PDB file obtained through Alphafold was uploaded to the ProteinPrepare tool. pKa calculation was performed at pH 4.0 to 7.0 for TvMP50 and TvGP63 and at pH 6.0 for TvCP2 without water molecules and ligands from the input PDB file. After the calculation, protonated PDB files were analyzed. The electrostatic surface properties were calculated using sequential calculation for multi-network focusing of automatic concentration in Adaptive Poisson-Boltzmann Solver (APBS).

The interaction of TvMP50 and TvGP63 with ligands (1,10-phenanthroline, phendione, Ag-phendione and Cu-phendione) and of TvCP2 with Cu-phendione were identified using the AutoDock Vina 1.1.2 program [25]. The predicted structures were prepared as described in the above, using pH 6.0, and subjected to molecular docking analysis. Auto DockTools (ADT) [26] was used to prepare the TvMP50, TvGP63, and TvCP2 file as a receptor. The atomic coordinates (3D) of the ligands were created using Discovery Studio v20 (Accelrys, San Diego, CA, USA)with the Chem3D-MM2 protocol applied for energy minimization (Chem3D Ultra, CambridgeSoft Co., 100 Cambridge ParkDrive, Cambridge, MA, USA, 02140), and the rigid root of each ligand was generated using AutoDockTools (ADT). The center of the gridbox in the TvMP50 center of mass was −0.101 × 0.399 × −1.521, while for TvGP63 it was −4.804 × 2.156 × 0.008 and for TvCP2 it was −2.691 × 0.922 × 18.558 on the x-, y-, and z-axes, respectively. The gridbox size (Å) was 90 × 60 × 80 for TvMP50, 110 × 100 × 80 for TvGP63, and 60 × 60 × 90 for TvCP2. The binding model with lowest free binding energy was researched from among ten different conformers for each ligand. Metallopeptidase and ligands complexes were visualized and analyzed using Discovery Studio, v20 (Accelrys, San Diego, CA, USA).

### 2.4. Proteolytic Activity Assay

*T. vaginalis* proteolytic activity was evaluated according to Weber et al. [27] using azocasein as substrate. After treatment, protein quantification was performed by the Coomassie blue method [28] and adjusted to a final concentration of 0.3 mg/mL. The assay was conducted in 0.1 M Tris-HCl, pH 7.0, for 90 min at 37 °C with 2% azocasein (Sigma-Aldrich, Co. St. Louis, CA, USA). The reaction was interrupted with 10% trichloroacetic acid and, after centrifugation at 10,000× *g* for 5 min, 1.8 N NaOH was added to the supernatants. Absorbance was measured at 420 nm and azocasein degradation of treated organisms was compared to control, with 100% peptidase activity. Furthermore, the cleavage over two specific peptide substrates, Z-Phe-Arg-AMC at 10 μM (Z-Phe-Arg 7-amido-4-methylcoumarin hydrochloride, Sigma-Aldrich) and DNP-Pro-Leu-Gly-Met-Trp-Ser-Arg at 25 μM (MMP2/MMP9 substrate, Sigma-Aldrich), was tested in order to evaluate the CP and MP activities, respectively. The substrates’ cleavage was evaluated in a spectrofluorometer (SpectraMax Gemini XPS, Molecular Devices, San Jose, CA, USA) using an excitation/emission wavelength of 280/360 nm for MP and 380/460 nm for CP. The reactions were initiated with the addition of 10 μg of protein and the proteolytic inhibitors tested were 1,10-phenanthroline (5 mM) and E-64 (*N*-*trans*-epoxysuccinyl-L-leucine-4-guanidinobutylamide) (10 μM).

### 2.5. Sodium Dodecyl Sulfate–Polyacrylamide Gel Electrophoresis (SDS-PAGE) Assay

The peptidase profile of *T. vaginalis* was assayed by gel electrophoresis containing 1% gelatin as proteinaceous substrate incorporated into the sodium dodecyl sulfate-polyacrylamide gel [17]. Trophozoites’ cellular protein was adjusted to 60 μg/mL of protein using the Coomassie method and then loaded in the slot [28]. After electrophoresis at a constant voltage (120 V) at 4 °C, 2.5% Triton X-100 was used to remove SDS. Afterwards, gel was added into digestion buffer comprising 50 mM phosphate buffer, pH 5.5, for 24 h at 37 °C. The gels were stained for 2 h with 0.2% Coomassie brilliant blue R-250 in methanol:acetic acid:water (50:10:40) and destained in a solution containing methanol:acetic acid:water (5:10:85). Low molecular mass standards (Sigma-Aldrich) were used for comparison of molecular masses.

### 2.6. Mass Spectrometry Analysis

The decrease in proteolytic activity modulation was observed in the gel, then the peptidases related to this region were determined by mass spectrometry analysis. The gel was digested as described by Martineli et al. [29]. The obtained peptides were submitted to reverse phase chromatography (Nano Acquity Ultra Performance LC-UPLC^®^ (Waters, Milford, MA, USA)) using a Nanoease C18, 75 μm ID at 35 °C. Elution occurred in a constant flow ramp of 0 to 60% acetonitrile in 0.1% trifluoroacetic acid at 0.6 nL/min. The analysis was performed on a G2-XS Q-TOF Xevo mass spectrometer^®^ and the identification was based on the genome database of *T. vaginalis* (*T. vaginalis* G3, G3; ATCC PRA-98, WGS project AAHC01000000) held on the *ProteinLynx Global Server* (PLGS) platform version 2.2.5 and Uniprot database (https://www.uniprot.org/, accessed on 9 November 2019).

### 2.7. Expression of mRNA through qRT-PCR

Evaluation of the relative gene expression of *T. vaginalis* peptidases was performed for the CPs (calpain, cathepsin L and B), MPs (TvMP50 and TvGP63), and aspartic-type peptidase (cathepsin D) [13,27]. Quantitative real-time PCR (qRT-PCR) was used to compare the mRNA levels expressed between trophozoites treated with Cu-phendione and the control without treatment. RNA extraction was performed with TRIzol [30] and analyzed with a Thermo Scientific NanoDrop 1000^®^ spectrophotometer, using only samples with adequate purity. To standardize the primer curves, we used cDNA produced through a High-Capacity cDNA Reverse Transcription Kit (Applied Biosystems). The analyses were performed according to Santos et al. [31] using the DNA topoisomerase II as reference gene. The GoTaq^®^ 1-step RT-qPCR reaction system was prepared with 5 μL of 2 GoTaq^®^ qPCR Master Mix, 0.2 μL of GoScript™ RT Mix, 0.1 or 0.2 μM of primer, and 2X μL RNA (5.0 ng). The conditions used for amplification and detection of the gene were activation of the enzyme at 95 °C for 10 min followed by 40 cycles composed of 95 °C for 10 s. The melting curve was performed by increasing the temperature from 60, 64, and 68 °C to 95 °C with 1 °C for 5 s. The peptidases, primer concentrations, and ideal annealing temperature are described in Table 1. Analysis of the relative expression of the genes was performed through Rotor-Gene Q series 2.1 software; the reference gene value (ΔCt) was used for threshold (Ct) cycles and compared to the control (ΔΔCt), with values expressed as 2^−ΔΔCt^.

### 2.8. Mammalian Cell LDH Release Assay

In order to evaluate the cytolysis capacity of *T. vaginalis* trophozoites pretreated or not with Cu-phendione, an assay was performed to detect lactate dehydrogenase enzyme (LDH) released by cells using Cyto Tox-One homogeneous membrane integrity assay (Promega, USA) reagent. Human vaginal epithelial lineage cells (HMVII) and VERO non-tumoral cell lineage were cultured in RPMI 1640 and DMEM medium, respectively, supplemented with 20% inactive bovine fetal serum and 100 μg/mL penicillin-streptomycin at 37 °C and 5% CO_2_. Prior to the experiment, 3 × 10^4^ cells were seeded in 96-well microplates and incubated until monolayer confluence. At this time, ATCC 30236, TV-LACM15, TV-LACM22, and TV-LACH4 were treated with Cu-phendione for 2 h at minimum lethal concentration, as described by Rigo et al. [18]. Afterwards, trophozoites were washed and suspended in RPMI or DMEM medium supplemented with 20% fetal bovine serum and its density adjusted to 5 × 10^5^ trophozoites/mL. The control went through the same process; however, only supplemented TYM medium was used during treatment. The cell medium was removed, then an aliquot of 100 μL of trophozoites was added to the monolayer and incubated for 6 h at 37 °C and 5% CO_2_. Afterwards, the LDH release was measured according to the manufacturers’ instructions. The background value refers to LDH released by cell medium supplemented with 20% inactive bovine fetal serum, while the maximum LDH release is related to cells exposed to 0.2% Triton X-100. The results were calculated using the following formula:$$\frac{\left(Experimental\;Value-background\right)}{\left(TritonX-100-background\right)}\times 100$$

### 2.9. Statistical Analysis

Experiments were carried out in triplicate and with at least three independent cultures (*n* = 3). The data were expressed by mean ± standard deviation. Statistical analysis was performed with GraphPad Prism 8.0.2 using Student’s *t-*test, and a significance level of 5% was applied to the data.

## 3. Results and Discussion

### 3.1. Docking Analysis Reveals That Cu-Phendione Interacts with the Active Site of TvMP50, TvGP63, and TvCP2

Considering that phendione has been shown to present activity against *T. vaginalis* cells both when free and when bound to copper or silver metal [18], these three compounds and a prototype MP inhibitor (1,10-phenanthroline) were used for computational analyses. Three-dimensional models of the enzymes were constructed using Alphafold, as represented in Figure 1, Figure 2, Figure 3 and Figure 4 (with the aim of evaluating the interaction of the compounds with MPs. The calculated structures have high prediction levels, with pIDDT (Local Distance Difference Test) results greater than 90% (Figure 1A) and 60% (Figure 2A). This test evaluates how well the environment of a reference structure is reproduced in a protein model by calculating the pairs of atoms in the former and the corresponding distance in the latter [32]. The structure of the modeled TvMP50 was submitted to active site identification analysis using Discovery Studio Visualizer v21. Prior to protein characterization analysis, its structure was protonated at pH 6.0 in order to mimic the characteristics of the enzyme in an in vitro environment. As shown in Figure 1B and Figure 2B, it was possible to identify the enzymatic catalytic site (ECS), represented in green. For TvMP50, the ECS was composed of three histidine residues at positions 215, 324, and 335. The amino acids found in the catalytic triad of the model obtained herein were in accordance with the description referring to the family of MP M24 [33]. At the ECS of TvGP63, a glutamic acid residue on position 208 and two histidine residues at positions 207 and 211 can be observed. Furthermore, it was possible to observe a difference in protonation of the enzyme with the increase in pH. As expected, titrable amino acids can change the load according to the alkalinization of the medium; however, the coupling capacity of the tested molecules did not change, demonstrating that pH change did not affect the inhibiting action of the test molecules. This result has great relevance, as the dysbiosis caused by the presence of pathogens in the vaginal environment is characterized by pH changes within the range evaluated in this experimental approach.

Subsequently, the interactions between the test compounds and the *T. vaginalis* MPs were investigated by molecular docking analysis. The results revealed that all compounds were able to bind to the active sites of both TvMP50 and TvGP63 MPs (Figure 3, Table 2 and Table 3). For TvMP50, Cu-phendione displayed the highest binding affinity (−9.7 kcal·mol^−1^), followed by Ag-phendione (−8.8 kcal·mol^−1^) and phendione (−6.1 kcal·mol^−1^). The same profile was obtained for TvGP63, Cu-phendione having the highest value (−10.7 kcal·mol^−1^) followed by Ag-phendione (−8.2 kcal·mol^−1^) and phendione (−6.5 kcal·mol^−1^). The classical MP inhibitor 1,10-phenanthroline showed a binding affinity of −6.0 kcal·mol^−1^ against both TvMP50 and TvGP63. It is interesting that the affinity value pattern demonstrated by docking with the MPs was the same as that observed in the anti-*T. vaginalis* action, in which Cu-phendione presented the highest activity against this protozoan [18]. It is relevant that both 1,10-phenanthroline and Cu-phendione showed interaction with histidine 335 from the catalytic triad (Figure 3D) of TvMP50; in addition, Cu-phendione presented interaction of the hydrogen bridge type with the active site (Table 2). All compounds showed interaction with both TvMP50 and TvGP63 active sites; however, Cu-phendione was chosen for the subsequent assays, as it presented the highest interaction (Figure 3E–H, Table 2 and Table 3). In addition, when analyzing the interaction of the Cu-phendione compound with TvCP2, it has a binding affinity of −8.7 kcal/mol. According to the anchoring position in the TvCP2 structure, the compound interacts with the amino acids TYR210 and VAL174 through hydrophobic interactions, with LYS312 and ASP171 through hydrogen bonds, and with ASP171 through electrostatic interactions (Figure 4, Table 4).

### 3.2. Cu-Phendione Inhibits the Peptidase Activities of T. vaginalis

Considering the interaction of Cu-phendione with the main trichomonad MPs, the effect of this compound in the enzymatic profile was evaluated further. Azocasein is one of the most reliable substrates for measuring proteolytic activity, as it has good color stability and no need for chromogenic reagents [34]. The same methodology used to characterize *T. vaginalis* peptidases has been used to evaluate the trichomonacidal effects of compounds such as 2,4-diamine-quinazoline derivative [27,35]. As shown in Figure 5A, trophozoites treated with Cu-phendione had a significant reduction (77.14%) in their ability to degrade the casein substrate compared to untreated parasite cells, demonstrating an important inhibitory effect on the peptidase pool.

In order to determine the effect of Cu-phendione on different peptidase classes produced by *T. vaginalis*, two fluorogenic peptide substrates were used; Phe-Arg was used to evidence the enzymatic activity of CPs, while MMP was used to measure MPs. As shown in Figure 5B,C, trophozoites treated with Cu-phendione showed a significant reduction in the ability to cleave both peptide substrates, suggesting impairment in the biological functions of these peptidases. An extensive review by Arroyo et al. [9] highlighted the main virulence properties related to CPs, such as cytoskeleton disruption, hemolysis, cytotoxicity, cytoadhesion, immunoglobulin degradation, and induction of host cell apoptosis. MPs are related to pathogenesis of *T. vaginalis* due to the cytotoxic action of these enzymes against host cells, in addition to the immunogenic activity found in the serum of male patients [10,12,36].

The significant reduction of MMP cleavage observed with Cu-phendione was comparable to 1,10-phenanthroline, a well-known MP inhibitor. Studies have already linked the action of Cu-phendione with inhibition of metallo-type enzymes from different clinically relevant microorganisms. Multifunctional elastase B, described as LasB, is an MP from *Pseudomonas aeruginosa* responsible for the degradation of extracellular matrix constituents, potentially leading to tissue injury. Galdino et al. [17] demonstrated that Cu-phendione can interact with the catalytic site of LasB, significantly inhibiting its enzymatic activity. Moreover, MP from *Leishmania* spp. proved to be a target of Cu-phendione, reducing both membrane-associated and secreted MP activities, with direct impacts on the establishment and maintenance of infection [19]. These results demonstrate an imperative effect of Cu-phendione on cellular homeostasis considering the different functions and MPs enrolled in the protozoal metabolism, leading to in-depth investigation of peptidases targeted by this compound.

### 3.3. Cu-Phendione Negatively Modulates Peptidase Production in T. vaginalis Cells

Electrophoresis was performed to identify the nature of enzymes with altered proteolytic activity after treatment. The zymogram in Figure 6 shows regions, with the different shades related to greater or lesser gelatin proteolysis. The dark regions refer to the presence of integral gelatin with greater staining by Coomassie after treatment with Cu-phendione. These regions were excised and analyzed by mass spectrometry and compared to the untreated control, where the enzymes had their proteolytic activity preserved. This methodology is commonly used in the investigation of peptidase activity from *T. vaginalis*. Cathepsin D-like aspartic peptidase, MPs, and several CPs have already had their proteolytic profiles identified by protein gels [13,37,38,39]. Indeed, this method has been shown to be suitable for comparing the effect of molecules with trichomonacidal action on protozoan peptidases [27].

In order to determine peptidases with reduced activity, the dotted region in the gel was extracted and analyzed by mass spectrometry. Through the UniProt database, 166 protein fragments were identified (Appendix A); those classified as peptidases according to the UniProt database are grouped in Table 5. TVAG_387200 was identified as MP-type GP63, while TVAG_193260 and TVAG_202060 were identified as CP-type ubiquitin hydrolase, previously identified in the degradome of *T. vaginalis.* Several families can be found in CP, where the CA clan is widely represented by proteins associated with deubiquitinating activity, in addition to the existence of a proteasome pathway linked to ubiquitin in various parasites [4]. Among the main deubiquitinating peptidases in protozoa, the C12 family are represented by ubiquitin terminal hydrolases, which mainly act in ubiquitin recycling when it is erroneously conjugated with intracellular nucleophiles, and the C19 family, which are ubiquitin-specific peptidases that oppose and co-evolve with ubiquitin E3 ligases [40].

### 3.4. Cu-Phendione Modulates the Expression of Peptidase Genes in T. vaginalis

Based on the results obtained in the evaluation of peptidase activity, this study investigated the effect of Cu-phendione treatment at the molecular level. To this end, analysis of the mRNA gene expression of the CP, MP, and aspartic peptidase families was performed, starting from the premise that protein synthesis can be activated or repressed under the influence of specific external agents [41]. After treatment with Cu-phendione, the relative expression of mRNA of the *calpain*, *cathepsin L*, *TvMP50*, and *TvGP63* genes showed significant reduction, with particularly emphasis in the *MP*-related genes. This profile could not be observed for expression of cathepsin types D and B (Figure 7).

In fact, Cu-phendione seems to have great importance for the expression of MPs. TvMP50 is part of the M24 family, known as methionyl aminopeptidase. The identification of this enzyme in the protozoan *T. vaginalis* occurred through the gene TVAG_403460, described as immunogenic only in the serum of male patients [11,36]. Another study confirmed the presence of mRNA of the TVAG_403460 gene in the isolate T016, referenced in the article by its access number in GenBank, JF263458.1, suggesting that this peptidase is involved in the cleavage of HOST mTOR [13]. The same gene was investigated by Puente-Rivera et al. [10], who demonstrated that the presence of zinc in the microenvironment can promote gene expression and elucidated its location in vesicular structures in the protozoan cytoplasm. In addition, the same study demonstrated that this MP is secreted in its active form and that both forms, secreted and endogenous, exert cytotoxicity against prostate cells, being considered a cytolytic effector involved in the pathogenesis of *T. vaginalis* [10]. In addition, the other MP investigated in this study, TvGP63, is representative of the M8 family. It is described as a zinc-dependent MP, and is not affected by classical inhibitors such as EDTA and phosphoramidon, while 1,10-phenanthroline is active against this MP [3]. Another study described 48 members of TvGP63 family in *T. vaginalis* and separated them into three main groups, -a/-b/-c, with amino acids preserved at position 81 and at positions 515 and 90, and amino acids highly conserved in 75% of TvGP63 sequences. The localization of TVAG_367130 was described in the plasma membrane, which has great relevance in cytotoxicity processes against cellular hosts [12]. In addition, the occurrence of the *GP63* gene in *T. vaginalis* has been demonstrated and referenced according to GenBank access number GU356538.1, demonstrating that MP acts in the cleavage of the mammalian target of rapamycin (mTOR) of the host cell [13].

### 3.5. Cu-Phendione Reduces the Cytolysis Induced by T. vaginalis in Mammalian Cells

Considering the importance of peptidases for *T. vaginalis* cytotoxicity, a cytolysis assay was performed to evaluate whether the pretreatment of trophozoites with Cu-phendione would have any effect in this virulence process. As demonstrated in Table 6, even 2 h of treatment with Cu-phendione was enough to lead to a significant reduction in parasite-mediated cytotoxicity, as observed in both clinical fresh and ATCC reference *T. vaginalis* isolates. In addition, the reduction of cytolysis after Cu-phendione treatment was registered by microscopy of all tested isolates and both mammalian cell lineages (HMVII and VERO) (Appendix A). These results demonstrate that the presence of Cu-phendione can modulate the virulence of *T. vaginalis*.

The *T. vaginalis* peptidases participate in several virulence processes against the host, which may vary according to more or less virulent phenotypic isolates. Heterogeneity cytolysis of human vaginal epithelial cells has been determined in different *T. vaginalis* isolates [42]. Arroyo et al. highlighted several virulence properties from CPs, and demonstrated that the cytoadherence and cytotoxicity processes present different levels among *T. vaginalis* isolates, as well as distinct proteolytic activity observed in zymogram profiles [9]. Despite being a complex process that involves the modulation of several pathways of the parasite, the key role of different peptidases in mediating the cytotoxicity process has been increasingly described in the literature. TvCP65 is involved in trichomonad host cellular damage through regulation, probably in the transcriptional and translational levels, with a direct link to the polyamine metabolism [43]. TvCP39 is a glycosylated CP located on the surface the parasite involved in trichomonad cytotoxicity [44]. In addition, MPs have been described as virulence factors. TvMP50 was identified in the cytoplasm and secretion products of trophozoites treated with zinc, and cytotoxicity toward prostatic DU145 cell monolayers was demonstrated [10]. The capacity of lucidin-ω-isopropyl ether to reduce the cytotoxicity of *T. vaginalis* against HeLa was related to the inhibition of TvMP50 proteolytic activity at the genetic level [45]. In this sense, the inhibition of peptidases by the compound investigated in the present study demonstrates a new option for development of novel anti-*T. vaginalis* agents.

## 4. Conclusions

Recently, our group demonstrated the effect of Cu-phendione in the oxidative metabolism of *T. vaginalis* due to the decreased function and gene expression of enzymes responsible for detoxification, leading to homeostasis imbalance. Moreover, the Cu-phendione complex led to parasite death by activating an apoptosis-like cell death pathway [20]. Data obtained in the present study suggest that these peptidases are not directly involved in the activation of apoptotic pathways; instead, the multiple targets of Cu-phendione can lead to disturbances in the proteolytic metabolism of the parasite. Moreover, while the compound causes the death of *T. vaginalis* trophozoites by disturbance of the oxidative and proteolytic metabolism, it additionally causes reduced toxic effects against human cells. Indeed, in the scenario where microorganisms are constantly adapting to develop new resistance mechanisms, the presence of a molecule that is able to alter parasite homeostasis by different pathways has a greater ability to circumvent the selective pressure caused by therapy. In this sense, Cu-phendione stands out as a potent antiparasitic molecule able to act in peptidases, with important inhibition capability against *T. vaginalis* MPs.

## Figures and Tables

**Figure 1 pathogens-12-00745-f001:**
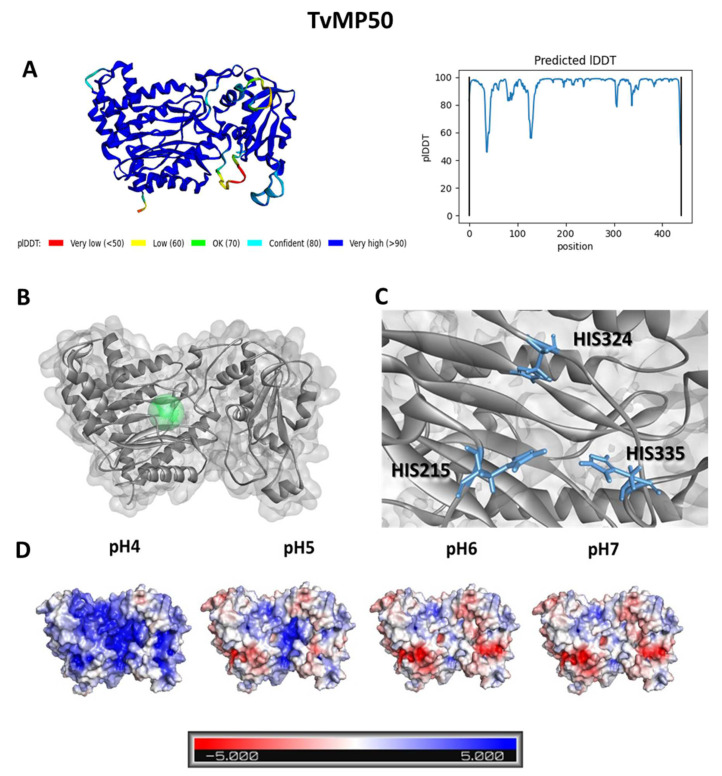
Three-dimensional model for TvMP50 of *Trichomonas vaginalis*. (**A**) PIDDT values for the amino acid sequence of the structure modeled using Alphafold. (**B**) Active site, represented by the green marking. (**C**) Identification of amino acids from the enzyme’s active site. (**D**) Electrostatic charge on the surface of the TvMP50 calculated for pH values of 4 to 7. The red-white-blue scale refers to the minimum (−5 kT/e, red) and maximum (5 kT/e, blue) potentials of the surface.

**Figure 2 pathogens-12-00745-f002:**
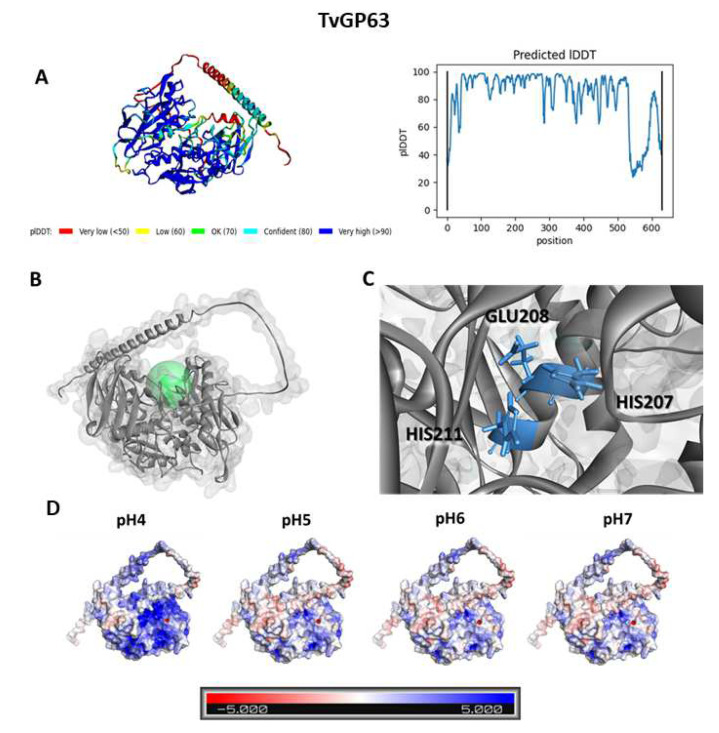
Three-dimensional model for TvGP63 of *Trichomonas vaginalis*. (**A**) PIDDT values for the amino acid sequence of the structure modeled using Alphafold. (**B**) Active site, represented by the green marking. (**C**) Identification of amino acids from the enzyme’s active site (**D**) Electrostatic charge on the surface of the TvMP50 calculated for pHs 4 to 7. The red-white-blue scale refers to the minimum (−5 kT/e, red) and maximum (5 kT/e, blue) potentials of the surface.

**Figure 3 pathogens-12-00745-f003:**
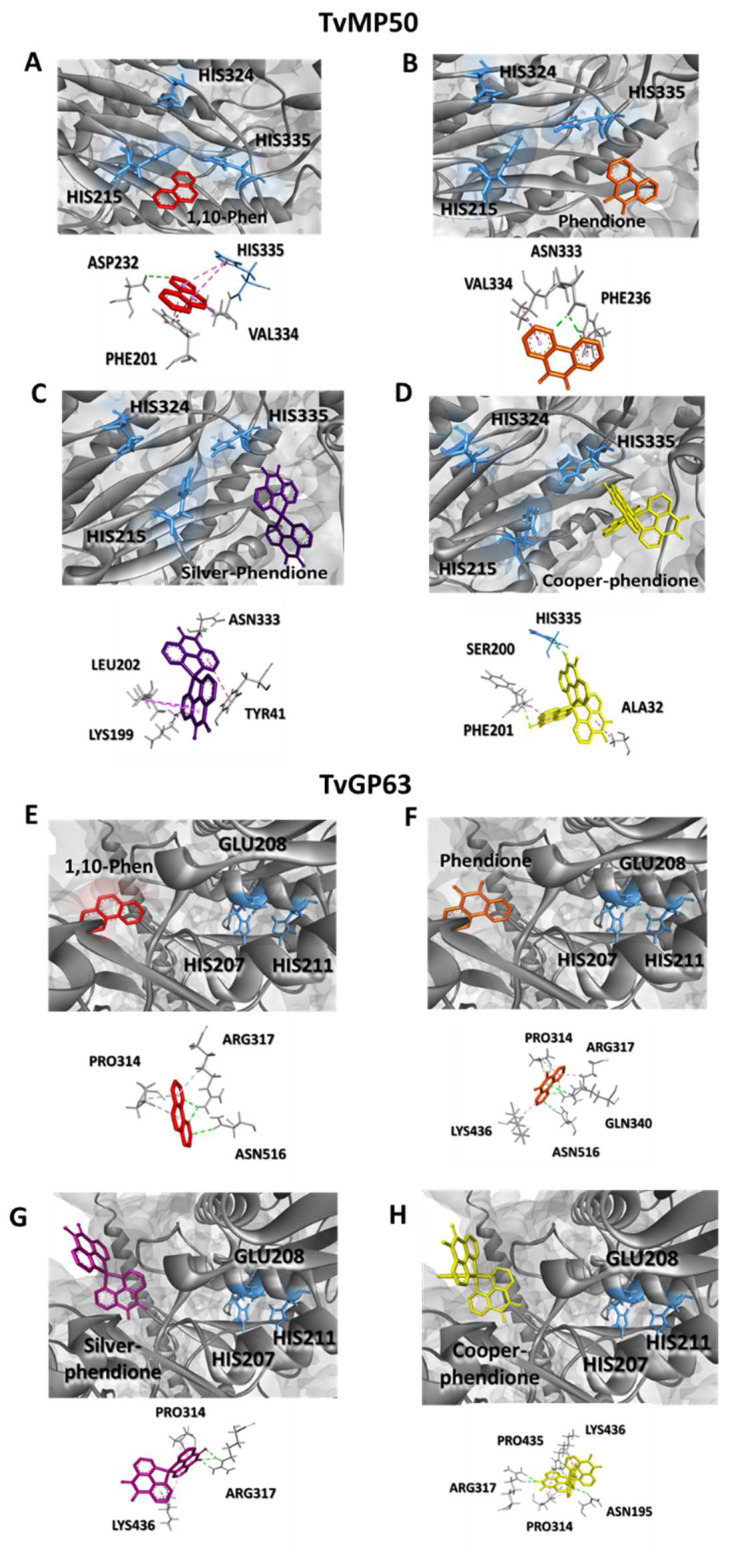
Docking pose with lower absolute energy (kcal·mol^−1^) and 3D interaction diagram of the compounds with the amino acids of TvMP50 and TvGP63 from *Trichomonas vaginalis*. (**A**,**E**) 1,10-phenanthroline, (**B**,**F**) phendione, (**C**,**G**) Ag-phendione, (**D**,**H**) Cu-phendione.

**Figure 4 pathogens-12-00745-f004:**
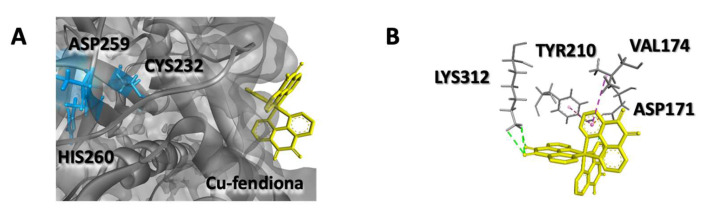
(**A**) Docking pose with lower absolute energy (kcal·mol^−1^) and (**B**) 3D interaction diagram of the Cu-phendione with the amino acids of TvCP2 from *Trichomonas vaginalis*.

**Figure 5 pathogens-12-00745-f005:**
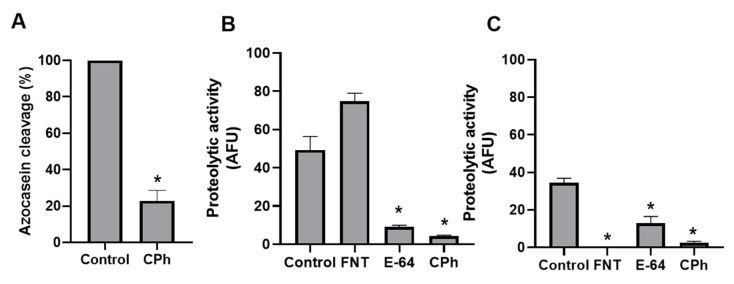
Cu-phendione inhibits proteolytic activity in *T. vaginalis*. (**A**) Azocasein cleavage rate by parasites treated (12.5 μM) or not with Cu-phendione (CPh), pH 7.0, 37 °C for 90 min. (**B**) Cleavage of Phe-Arg substrate by parasite lysate at pH 5.0, 37 °C for 1 h. Results are expressed as arbitrary fluorescence units (AFU). (**C**) Cleavage of DNP-Pro-Leu-Gly-Met-Trp-Ser-Arg (MMP) substrate by parasite lysate at pH 9.0, 37 °C for 1 h. Results are expressed as arbitrary fluorescence units (AFU). CTL: Untreated control. FNT: 1,10-Phenanthroline. E-64: *L-trans*-epoxysuccinyl-L-leucylamido(4-guanidino) butane. (*) indicates statistically significant decrease (Student’s *t-*test; *p* < 0.05).

**Figure 6 pathogens-12-00745-f006:**
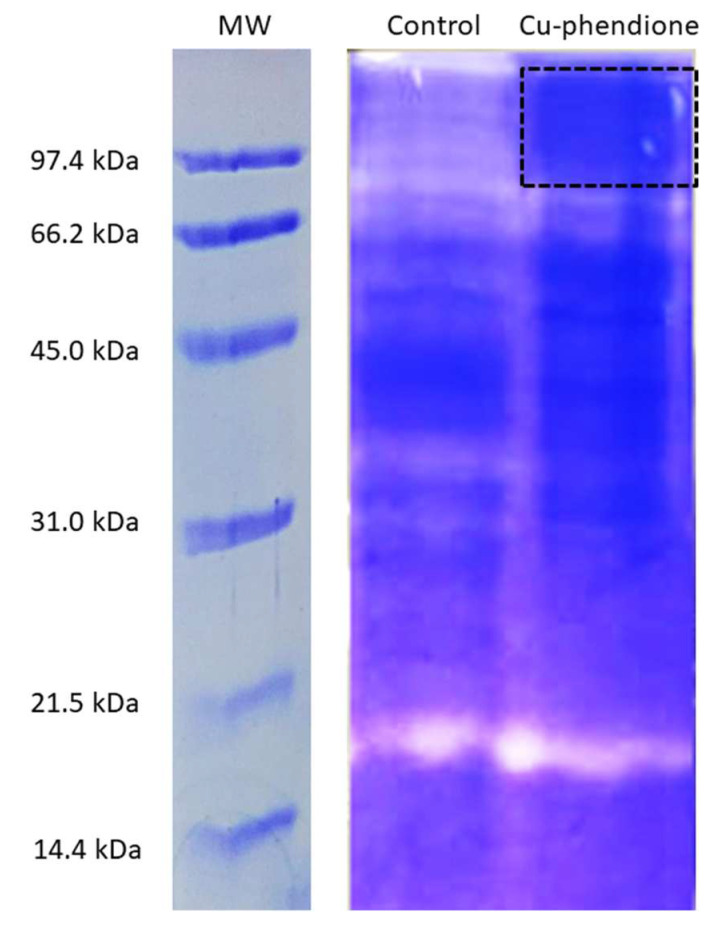
Zymogram of the effect of Cu-phendione on *T. vaginalis* peptidases. Peptidase zymograms with gelatin after 24 h of incubation of ATCC 30236 isolate treated or not with Cu-phendione at MIC (12.5 μM). The region marked by the square was cut out of the zymogram for gel digestion. MW: Molecular weights in SDS-PAGE without gelatin.

**Figure 7 pathogens-12-00745-f007:**
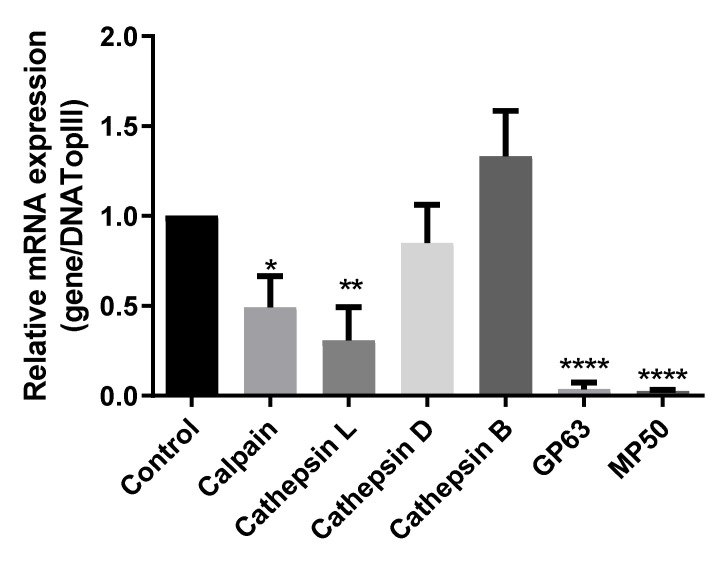
Analysis of the mRNA expression of calpain, cathepsin L, cathepsin D, cathepsin B, Gp63, and MP50 relative to the reference gene (*DNATopII*) in *T. vaginalis* treated with Cu-phendione. Statistically significant differences: (*) *p* < 0.05; (**) *p* < 0.005; (****) *p* < 0.0001.

**Table 1 pathogens-12-00745-t001:** Parameters used for qRT-PCR of *Trichomonas vaginalis* peptidases.

Identification	Primers (5′-3′)	Primer Concentration	Annealing Temperature (°C)
**Calpain** **(TVAG_161170)**	F: CCA AAG GAT GCA CGA ATT TT R: GCG GAC ATT AGC TGG TTT GT	0.2	60
**Cathepsin L (TVAG_057000)**	F: CTG AAC TCG CTA AGG CTG CT R: GTT GCG GAC GAT CCA GTA GT	0.1	68
**Cathepsin D (TVAG_336300)**	F: TAC AAC CCA GAT GCT TTC TC R: GCA TAG ATG TGC CTG TAT CA	0.2	64
**Cathepsin B (TVAG_488380)**	F: CAA GAG TGC GGT TGT TGC TA R: TAA GCG CAT GCT GTT CAA GA	0.2	60
**TvGP63 (GU356538.1)**	F: ACG CTG TCC TTG CAA TTC TT R: TTG CGT TTT CTT TTG TGC AT	0.1	60
**TvMP50 (TVAG_403460)**	F: TCT CGA CTG CGG ATT CTT CT R: TCC GAC GTG ATG AGT CAA AC	0.1	64

**Table 2 pathogens-12-00745-t002:** Docking results predicted by AutoDock Vina: binding affinity, interaction type, amino acid residue, and geometric distance (Å) for TvMP50 of *Trichomonas vaginalis* with test compounds.

Compounds	Binding Affinity (kcal·mol^−1^)	Interaction Type	Amino Acid	Distance (Å)
1,10-Phenanthroline	−6.0	Hydrogen bonding	ASP232	3.47
		Hydrophobic	PHE201	4.02
				4.81
			HIS335	4.98
				5.46
			VAL334	4.87
				5.22
Phendione	−6.1	Hydrogen bonding	ASN333	2.02
				2.46
		Hydrophobic	PHE236	4.62
			VAL334	5.18
Ag-Phendione	−8.8	Hydrogen bonding	ASN333	1.96
				2.99
		Hydrophobic	TYR47	5.22
			LEU202	5.24
			LYS199	5.37
			LEU202	4.68
Cu-phendione	−9.7	Hydrogen bonding	HIS335	2.46
			PHE201	2.49
		Hydrophobic	SER200/PHE201	4.42
			ALA32	5.29

**Table 3 pathogens-12-00745-t003:** Docking results predicted by AutoDock Vina: binding affinity, interaction type, amino acid residue, and geometric distance (Å) for TvGP63 of *Trichomonas vaginalis* with test compounds.

Compounds	Binding Affinity (kcal·mol^−1^)	Interaction Type	Amino Acid	Distance (Å)
1,10-Phenanthroline	−6.0	Hydrogen bonding	ARG317	2.50
				2.37
			ASN516	3.17
		Hydrophobic	PRO314	4.10
				4.84
			ARG317	4.99
Phendione	−6.5	Hydrogen bonding	ARG317	2.53
				2.43
			ASN516	3.13
			GLN340	3.31
		Hydrophobic	PRO314	5.03
				4.15
			ARG317	4.98
			LYS436	5.44
Ag-Phendione	−8.2	Hydrogen bonding	ARG317	2.58
				2.62
				1.80
		Hydrophobic	LYS436	5.00
				5.49
			PRO314	5.18
				4.25
				5.18
Cu-phendione	−10.7	Hydrogen bonding	ARG317	2.47
				2.13
			ASN195	3.23
			LYS436	3.04
		Hydrophobic		4.40
				4.98
			PRO435	5.41
			PRO314	4.81
				4.83
				4.05
				4.89

**Table 4 pathogens-12-00745-t004:** Docking results predicted by AutoDock Vina: binding affinity, interaction type, amino acid residue, and geometric distance (Å) for TvCP2 of *Trichomonas vaginalis* with test compounds.

Compounds	Binding Affinity (kcal·mol^−1^)	Interaction Type	Amino Acid	Distance (Å)
Cu-phendione	−8.7	Hydrogen bonding	LYS312	2.88
				2.99
				2.34
			ASP171	3.36
		Electrostatic		3.35
		Hydrophobic	TYR210	5.41
			VAL174	5.37

**Table 5 pathogens-12-00745-t005:** Peptidases from *Trichomonas vaginalis* found in the band demarcated in the zymogram (Appendix A) and identified by mass spectrometry.

PGLS ^a^	UniProt		
Protein Code	Gene	Identification	Molecular Weight
A2G9D4	TVAG_387200	GP63-like	58.572
A2DH40	TVAG_193260	ubiquitin hydrolase-like cysteine peptidase	179.696
A2DWM7	TVAG_202060	Ubiquitin (carboxy-terminal) hydrolase-cysteine peptidase	254.796

^a^ Protein Lynx Global Server (PLGS).

**Table 6 pathogens-12-00745-t006:** LDH release by the HMVII and VERO cell lineages after contact with *Trichomonas vaginalis* ^a^.

Treatment Condition	LDH Release (%)	
	HMVII	VERO
**Triton X-100**	100	100
**Tv ATCC 30236**	3.8 ± 0.033	7.4 ± 0.217
**Tv ATCC 30236 + CPh**	0 ± 0.004	0 ± 0.030 *
**TvLAC-M22**	8.8 ± 0.03	28.6 ± 0.047
**TvLAC-M22 + CPh**	0 ± 0.01 *	0 ± 0.061 *
**TvLAC-M15**	9.4 ± 0.010	29.6 ± 0.238
**TvLAC-M15 + CPh**	0 ± 0.009 *	0 ± 0.113 *
**TvLAC-H4**	6.7 ± 0.012	29.3 ± 0.010
**TvLAC-H4 + CPh**	0 ± 0.03 *	0 ± 0.026 *

^a^ ATCC 30236, TVLAC-M22, TVLAC-M15 and TV-LAC-H4 isolates treated or not with Cu-phendione (CPh). The data were expressed as a percentage of total lysis, considering Triton X-100 treatment as positive control (100% of LDH release). The data were analyzed by Student’s *t*-test, where (*) *p* < 0.05.

## Data Availability

All of the data generated during the current study are included in the manuscript and/or the Appendix A.

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
