# Peer review of "Peptidases Are Potential Targets of Copper(II)-1,10-Phenanthroline-5,6-dione Complex, a Promising and Potent New Drug against Trichomonas vaginalis"

_pathogens, 2023, doi:10.3390/pathogens12050745_

Round 1
Reviewer 1 Report
The work by Rigo et al entitled: “Peptidases are Potential Targets to Copper(II)-1,10-Phenanthro- 2 line-5,6-dione Complex, a New Promising and Potent Drug against 3 Trichomonas vaginalis” evaluated the influence of Cu-phendione on the modulation of proteolytic activities produced by T. vaginalis by biochemical and molecular approaches. Cu-phendione showed a strong inhibitory potential against T. vaginalis peptidases, especially cysteine- and metallo-type peptidases. The latter revealed a more prominent effect at the transcript and proteolytic activity levels. Molecular Docking analysis confirmed the interaction of Cu-phendione with the active site of both TvMP50 and TvGP63 metallopeptidases, with high binding energy (-9.7 and -10.7 kcal.mol-1, respectively). In addition, Cu-phendione significantly reduced the trophozoite-mediated cytolysis in human vaginal (HMVII). However, it was not so clear with monkey kidney (VERO) epithelial cell lineages. These results highlight the antiparasitic potential of Cu-phendione, acting by interacting with important T. vaginalis virulence factors.
The research presented here is very interesting and quite deep towards the molecular mechanism of a potential trichomonacidal compound. However, some issues need to be addressed to accept the manuscript.
Major issues:
1. Pag. 14. There are some results shown in Figure 4B and C that need to be explained and modified. For example:
a) Fig. 4B: Why the CP proteolytic activity of Tv lysate treated with FNT showed greater AFU than the untreated control?
b) Please explain how the CPh compound could inhibit CP proteolytic activity even better than E-64 inhibitor?
c) Please include a docking analysis using the TvCP2 Alphafold predicted 3D structure and the Cu-phendione compound to support the experimental data obtained in the inhibition enzyme activity assays.
d) Fig. 4C. Please explain why and how the E-64 partially inhibited the MP activity of Tv lysates.
e) Please use the same scale for AFU in Fig. 4C as in Fig. 4B
2. Pag. 16. Figure 6. There are some results shown in this figure that need to be explained and complemented.
a) It is recommended that the CPh inhibitor be tested in an inhibition kinetic reaction using human Cathepsin L commercial enzyme to demonstrate that the effect of this compound is over the proteolytic activity directly and not a consequence of reduction on the CP transcript, as shown in Fig. 6?
f) How do you explain the effect of this compound on the amount of transcript on different type of proteolytic enzymes, as shown in Fig. 6??
3. Pag. 17 Table 5. Please check the % shown for LDH release for each Tv isolate with both cell lines. Appears to be no coincidence with the images shown as suppl. Fig. S1 and S2 that displayed major monolayer destruction by parasites without treatment. It seams that they should be 10x more ie. 38, 88, 94, 67% instead of those shown as 3.8, 8.8, 9.4, 6.7 for HMVII cytolysis in the absence of CPh. The Vero cell monolayers appear no to be well protected by the CPh parasite treatment. At least no clear monolayers is appreciated.
Line 268. Please check the formula to calculate the percentage of LDH release. In the manufacturer’s instructions include to multiply by 100.
Some of the images showed multiple parasites remaining in the wells. Can these alter the measurement of LDH release?
Minor issues:
Misspelling, typos, extra words, or confusing sentences need to be corrected throughout the text. For example
Lines 106, 112. The references still have the author’s last name.
Line 110: “…to… need to be removed.
Line 137: “…blue…trypan blue… Please correct it.
Line 148: “MIC” needs to be define the first time being used.
Line 168: …the crystalline structures. Please correct it. i.e. “…for 3D modeled structures …”
Line 178: “…with lowest…” Please change to “…with the lowest…”
Lines 208-209. This sentence is confusing. Please modify it. Thank you.
Lines 227-228: Please correct “catepsin” for “cathepsin”.
Lines 259-261. Tv suspended in RPMI or DMEM medium with 20% FBS. Does the serum or medium composition (pyruvate presence) interfere with cytolysis measured by the LDH release?
Line 312: It should be TvGP63 instead of TvMP50. Please correct it.
Line 344: Is a missing word in the sentence such as “substrate”. Please check it.
Line 493: “HELA” It should be “HeLa”?? Please correct it as need it.
Line 510: Please mention the evidence supporting that “Cu-phendione acts in peptidase at posttranscriptional level”. Thank you.
Reviewer 2 Report
Peptidases are Potential Targets to Copper(II)-1,10-Phenanthroline-5,6-dione Complex, a New Promising and Potent Drug against Trichomonas vaginalis
The study of Rigo et al. has an excellent execution in the molecular modeling against Trichomonas vaginalis antiparasitic activity and the health problems it represents. The article is well-detailed in the in vitro and in silico tests.
In general, some words/paragraphs are minor size letters. Please, revise all document sections and unify them.
Line 106, 110 &112. Remove the citation style like “Lima 2021” and “Liveira 2023”.
Line 208. °C, it seems, in another format.
Line 240 & 241. Abbreviation of seconds must be in “s,” not “sec.”
Line 268. Place formula 1 in the following line.
Line 269. The authors do not write the Software and version for statistical analyses.
Line 400. In Figure 5, the word “peptidases” has another letter size.
Line 433. Figure 6 legend has another letter size.
Line 472. Table 5, the authors should remove the lines inside the table. In addition, authors should express each percentage with the ±SD values.
Line 512. Remove the period from the subtitle.
Reviewer 3 Report
In this study, the effect of Cu-phendione on the modulation of proteolytic activities produced by T. vaginalis was investigated.
Cu-phendione showed a strong inhibitory potential especially against metallo-type peptidases and cysteine peptidases.
The interaction of Cu-fendione with the active site of both TvMP50 and TvGP63 metallopeptidases with high binding energy has been demonstrated.
Cu-phendione emerges as a potent antiparasitic molecule with significant inhibition of T. vaginalis MPs.
1. The main question addressed by the research is the effectiveness of Cu-phendione Peptidases against Trichomonas vaginalis
2. The topic is unique as it determines the activity of Copper-phendione. It fills this gap as it explores new and potential therapeutic agents on T. vaginalis in the field.
3. Compared to other published material, the topic clearly reveals the discovery of a powerful new agent
4. No need for special improvements with respect to the methodology
5. The results are consistent with the evidence and arguments presented. addresses the main question asked. Treatment with copper fendione inhibits the proteolytic activity of Trichomonas vaginalis. causes decreased vaginal epithelial cytolysis, and has emerged as a Promising New and Powerful Drug.
6. References are appropriate
7. Please, tables and figures are sufficient. The figures are especially descriptive and clear.
Reviewer 4 Report
In this manuscript, the authors continue the study related to the trichomonacidal effect of the 1,10-phenanthroline-5,6-dione complex. In previous studies this research group confirmed the effect against T. vaginalis. In this research work they focus on evaluating its effect against parasite proteolytic enzymes using a large panel of biochemical and in silico determinations.
In this research work the authors focus on evaluating its effect against parasite proteolytic enzymes using a large panel of biochemical and in silico determinations.
I find the topic relevant in this field as the experiments carried out are aimed at determining whether the mechanism of action is related to the alteration in the behavior of certain peptidases of the parasite.
This reviewer considers that the experiments were carried out following rigorous methods with adequate controls.
This paper adds, to the subject area, novel strategies and methods to evaluate potential modes of action
This reviewer considers that the experiments were carried out following rigorous methods with adequate controls.
The conclusions are consistent with the results obtained
The references are adequate
Nevertheless, this reviewer has only one suggestion to make to the authors:
In Figure 4, section C, the full name of MMP could be indicated.
Reviewer 5 Report
This paper evaluated the influence of Cu-phendione on the modulation of proteolytic activities produced by T. vaginalis by biochemical and molecular approaches. Cu- phendione showed a inhibitory activity against cysteine- and metallo-type peptidase in T. vaginalis. Metallo- peptidase revealed a effect at post-transcriptional levels. Cu-phendione interacts with the active site of metallopeptidases both TvMP50 and TvGP by molecular structure expectation. Cu-phendione significantly reduced the trophozoite-mediated cytolysis in host epithelial cells
Comment
Overall, the experimental procedure to reveal the virulent proteases and potential drug target of Trichomonas is innovative. However, this referee pointed out several statements in results section and controls in Figures are missing.
Authors cutoff more than 100 kDa bands (lane 402) from Zemogram in Figure 5. Then, authors showed three proteases among 166 identified protein (lane 407, data not shown) in Table4 identified from mass -spectrometry analysis. The referee pointed that all three proteins listed in Table 4 are all different molecular mass from 100kDa.
That is, TVAG_387200 (GP3-like) is a 516 amino acid residue with expected 58 kDa protein.
TVAG_193260 (ubiquitin hydrolase-like cysteine peptidase ) is a 179 amino acid residue with expected 179 kDa protein
YVAG_202060 (ubiquitin hydrolase-like cysteine peptidase ) is a 254 amino acid residue with expected 254 kDa
Author should be described why discrepancy exist between molecular weight of proteases shown is Table 4 and Figure 5.
Author should show all the protein list in supplemental table (166 protein fragment), show how many proteins with molecular mass around 100 kDa was identified, and ranked proteins by obtained peptide numbers to revise Table 4.
Figure 6
This mRNA gene expression analysis of RT-PCR doesn’t make sense to this referee. Why authors did not focus on novel proteases identified in Table4 for the RT-PCR, but focus on GP63 and MP50? How does Cu-phendione regulate the protease mRNA expression? Are they intercorporate into the nucleus to modulate gene transcription? Are GP63 and MP50 mRNA repress under the treatment of Cu-phendione as well as protease activity inhibited?
Which is the white bands corresponding to GP63 and MP50 in Zymogram assay? Around 60 kDa, the referee hard to see the specific bands corresponding to GP63 and MP50.
Table 5
Lanes 464-466, authors described “As demonstrated in table 5, only 2 h of treatment with Cu-phendione were enough to lead to a significant reduction in parasite-mediated cytotoxicity,”, however authors treated Cu-phendione for 24h (Fig 5, lane 401) or 1h (Fig 4, lane 381). Which treatment is the best to examine the effect on Cu-phendione?
Authors showed average value of LDH release from three independent assay in Table 5. Please add standard deviation in the value. Describe which values are compared to calculate the Student's t-test shown in asterisk in Table 5.
Round 2
Reviewer 1 Report
The authors have consider all comment and observations and responded point by point to each one of them.
Reviewer 5 Report
In the cover letter to revised manuscript, author should be include manuscript page number and line numbers, which description was edited in the revised manuscript. It makes easy for the referee's understanding.